# Model-based evaluation of cloud geometry and droplet size retrievals from 2-D polarized measurements of specMACS

Lea Volkmer[1], Veronika Pörtge[1], Fabian Jakub[1], and Bernhard Mayer[1]

[1]Meteorologisches Institut, Ludwig-Maximilians-Universität München, Munich, Germany

**Correspondence:** Lea Volkmer (L.Volkmer@physik.uni-muenchen.de)

**Abstract.** Cloud radiative properties play a significant role in radiation and energy budgets and are influenced by both the cloud top height and particle size distribution. Both cloud top heights and particle size distributions can be derived from two-dimensional intensity and polarization measurements by the airborne spectrometer of the Munich Aerosol Cloud Scanner (specMACS). The cloud top heights are determined using a stereographic method (Kölling et al., 2019) and the particle size distributions are derived in terms of the cloud effective radius and the effective variance from multidirectional polarized measurements of the cloudbow (Pörtge et al., 2023). In this study, the accuracy of the two methods is evaluated using realistic 3-D radiative transfer simulations of specMACS measurements of a synthetic field of shallow cumulus clouds and possible error sources are determined. The simulations are performed with the 3-D Monte Carlo radiative transport model MYSTIC (Mayer, 2009) using cloud data from highly resolved LES simulations. Both retrieval methods are applied to the simulated data and compared to the respective properties of the underlying cloud field from the LES simulations. Moreover, the influence of the cloud development on both methods is evaluated by applying the algorithms to idealized simulated data where the clouds did not change during the simulated overflight of one minute over the cloud field. For the cloud top height retrieval an absolute mean difference of less than $70\,\mathrm{m}$ with a standard deviation of about $130\,\mathrm{m}$ compared to the expected heights from the model is found. The elimination of the cloud development as a possible error source results in mean differences of $(46 \pm 140)\,\mathrm{m}$. For the effective radius, an absolute average difference of about $(-0.2 \pm 1.30)\,\mathrm{\mu m}$ from the expected effective radius from the LES model input is derived for the realistic simulation and $(-0.03 \pm 1.28)\,\mathrm{\mu m}$ for the simulation without cloud development. The difference between the effective variance derived from the cloudbow retrieval and the expected effective variance is $(0.02 \pm 0.05)$ for both simulations. Additional studies concerning the correlations between larger errors in the effective radius or variance and the optical thickness of the observed clouds revealed that low values in the optical thickness do not have an impact on the accuracy of the retrieval.

## 1 Introduction

On average, clouds cover about $67\,\%$ of the Earth's surface (King et al., 2013) and therefore, largely impact the global radiation and energy budgets determining our climate. With regard to the Earth's energy budget, clouds have both a cooling and a warming effect. On the one hand, the cooling effect originates from the reflection of the incoming shortwave radiation from the sun back to space which is determined by the optical properties of the clouds. Those optical properties depend on the

phase of the cloud (pure liquid or ice or mixed-phase respectively) and the shape and size of its particles. On the other hand, clouds absorb longwave radiation originating from the Earth's surface while emitting at lower temperatures which results in a greenhouse effect in the atmosphere. Since temperature decreases with height in the Earth's troposphere, the greenhouse effect increases with cloud top height. Therefore, the exact knowledge of the cloud top height is important to determine the impact of clouds onto the longwave radiation budget of the Earth.

As clouds form almost anywhere around the globe and appear in a large variety of cloud types (from optically thick cumulonimbus to thin cirrus clouds), their exact characterization is important to resolve their impact both on our daily weather as well as on the long-term climate. But resolving clouds in numerical weather prediction and climate models is limited due to their high spatial and temporal variability requiring high computational costs. Thus, most models rely on cloud parametrizations which are often based on measurement studies making the observational characterization of clouds important (e.g., Martin et al., 1994; Seifert and Beheng, 2001, 2006).

In recent decades, much effort has been made to better understand clouds and their feedback mechanisms to climate change, both from the modeling and observational sides. In particular, it has been accomplished to reduce the uncertainty in the global cloud feedback which is now likely expected to be positive with a high confidence as indicated by the most recent IPCC report (Forster et al., 2021). This was achieved by evaluating the regional feedbacks of clouds separately. For this, extensively studying the interaction between clouds, circulation and climate is indispensable (Bony et al., 2015). Airborne field campaigns such as the Next-Generation Aircraft Remote Sensing for Validation (NARVAL-I and II; Stevens et al., 2019) and the EUREC[4]A (Elucidating the role of clouds-circulation coupling in climate) field campaign (Bony et al., 2017) took place in the vicinity of Barbados to classify the macro- and microphysical properties of trade-wind cumuli. Other campaigns, such as the Arctic Cloud Observations Using Airborne Measurements during Polar Day (ACLOUD) and Physical Feedbacks of Arctic Boundary Layer, Sea Ice, Cloud and Aerosol (PASCAL) (both described in Wendisch et al., 2019) as well as the recent HALO-(AC)[3] campaign (Arctic Air Mass Transformations During Warm Air Intrusions and Marine Cold Air Outbreaks) were conducted for the characterization of clouds in the Arctic and their role in the Arctic Amplification. During the NARVAL expeditions, EUREC[4]A and HALO-(AC)[3] the German research aircraft HALO (High Altitude and LOng range research aircraft; Krautstrunk and Giez, 2012) was operated as a cloud observatory (Stevens et al., 2019). On board of HALO, the spectrometer of the Munich Aerosol Cloud Scanner (specMACS; Ewald et al., 2016) provides wide-field and spatially highly resolved radiance measurements from which both, cloud top heights and cloud optical properties can be obtained. The instrument consisted originally of two hyperspectral line cameras covering the wavelength range between $400\,\mathrm{nm}$ and $2500\,\mathrm{nm}$ (Ewald et al., 2016) but has been extended by two polarization resolving RGB cameras (Phoenix 5.0 MP Polarization Model) prior to the EUREC[4]A campaign (Pörtge et al., 2023). The wide combined field of view of about $90\,^{\circ} \times 120\,^{\circ}$ of the two cameras allows deriving cloud top heights and cloud droplet size distributions for a large area from spatially highly resolved intensity measurements at resolutions of $10-20\,\mathrm{m}$ at usual flight altitudes of $10\mathrm{km}$ (Pörtge et al., 2023). Moreover, the cameras provide simultaneous measurements at a framerate of $8\,\mathrm{Hz}$ resulting in a high temporal resolution. With the intensity measurements of the two RGB cameras, the cloud top heights are derived using a stereographic reconstruction method of the cloud geometry described by Kölling et al. (2019) for data of the 2-D RGB camera installed prior to the EUREC[4]A campaign. To summarize, the algorithm relies on

the identification of points on the cloud surface using contrast gradients. The reidentification of detected points in subsequent images and the associated observation from different perspectives enables localization in 3-D space. The polarization measurements allow to determine cloud droplet size distributions in terms of the effective radii and effective variances of liquid water clouds derived from observations of the cloudbow (Pörtge et al., 2023). Hereby, the dependency of the polarized scattering phase function of water clouds in the scattering angle range between $135\,^\circ$ and $165\,^\circ$, the region of the cloudbow, on the size distribution of the cloud droplets is used to determine effective radius and variance of the observed cloud targets. The cloud targets are defined as clusters of $10 \times 10$ cloudy pixels, and thus, have an approximate size of $100\,\mathrm{m} \times 100\,\mathrm{m}$ depending on the actual distance to the cloud. The cloud targets are observed from multiple viewing angles in subsequent images while flying over the clouds. The retrieval is described in Pörtge et al. (2023) and consists of three steps which are shortly summarized in the following. First, possible cloud targets which are observed in the cloudbow region are identified. Then, the identified cloud targets are located in 3-D space using the corresponding cloud top heights from the stereographic reconstruction algorithm. Hence, an accurate determination of the cloud top heights is important as small errors in the cloud top height will lead to large localization errors of the cloud targets in subsequent images. In case of inaccurate cloud top height data, the cloudbow signal will be wrongly aggregated which in turn leads to errors in the derived cloud droplet size distribution. This further motivates the accuracy assessment of the cloud top height retrieval performed in this paper. Finally, pre-calculated polarized scattering phase functions from Mie theory are fitted against the polarized radiance measurements and the best fit determines the effective radius and variance of the cloud target.

Similar techniques have been successfully applied to several space- and airborne instruments, such as POLDER (Polarization and Directionality of the Earth's Reflectances; Deschamps et al., 1994; Bréon and Goloub, 1998; Bréon and Doutriaux-Boucher, 2005; Shang et al., 2015), RSP (Research Scanning Polarimeter; Cairns et al., 1999; Alexandrov et al., 2012a), AirHARP (Airborne Hyper-Angular Rainbow Polarimeter; Martins et al., 2018; McBride et al., 2020) and AirMSPI (Airborne Multiangle SpectroPolarimetric Imager; Diner et al., 2013; Xu et al., 2018).

As both, cloud top heights and cloud optical properties determine the radiative properties of clouds and their feedback with regard to climate change, it is important to accurately measure those properties. In this study, the benefits of realistic 3-D radiative transfer simulations generated with the radiative transfer model MYSTIC (Monte Carlo code for the physically correct tracing of photons in cloudy atmospheres; Mayer, 2009) are exploited to evaluate the retrieval results and determine their accuracies. To do so, the usage of simulations is important as they rely on fully self-consistent cloud and radiation fields while for example comparisons to other instruments always depend on the different sensitivities. Moreover, it is often hard to find suitable measurements and even for large measurement campaigns such as EUREC[4]A with coordinated flights of remote sensing and in situ aircraft, simultaneous measurements of the same cloud and in particular its cloud top are rare. Furthermore, model simulations allow separating the different error sources since one has control over all model variables. For example, the investigation of the influence of the cloud development during the aircraft overpass is possible by assuming either a realistically evolving or a temporally constant cloud field.

The benefits of radiative transfer simulations for the accuracy assessment of cloud droplet size retrievals have also been used by Alexandrov et al. (2012a), who performed various tests on simplified 1-D and realistic 3-D radiative transfer simulations of

polarized reflectance measurements of the RSP instrument. For example, it was studied how aerosol layers of different optical thicknesses above the cloud layer or the presence of multiple cloud layers affect the RSP retrieval. Although the retrieval algorithms for RSP and specMACS are based on the same principals, namely multi-angle observation of the cloudbow and the fit of the polarized phase functions to the observations, the retrievals differ from each other because of the different properties of the instruments: While RSP is an along track scanning instrument with defined viewing angles and nine spectral channels (Cairns et al., 1999), the polarization cameras of specMACS measure 2-D images (Weber et al., 2023) which allows the retrieval of the cloud droplet size distribution for broader parts of the clouds. However, this comes at the cost of broader spectral response functions which might influence the accuracy of the polarimetric cloud droplet size distribution retrieval and will be tested with this work. In contrast to Alexandrov et al. (2012a) who mainly used the $865\,\mathrm{nm}$ wavelength for the simulations, we applied the full spectral response functions of the cameras with central wavelengths (bandwidths) of approximately $620\,\mathrm{nm}$ ($66\,\mathrm{nm}$), $546\,\mathrm{nm}$ ($117\,\mathrm{nm}$) and $468\,\mathrm{nm}$ ($82\,\mathrm{nm}$) (Pörtge et al., 2023). Further, the whole retrieval procedure including the identification of possible cloud targets and the geolocalization based on the stereographic cloud top heights will be tested.

The polarimetric cloudbow retrieval was further studied in Miller et al. (2018). In this work, Large-Eddy Simulations (LES) are used in combination with 1-D radiative transfer simulations for the comparison of cloud droplet size distributions derived from the bispectral MODIS (Moderate Resolution Imaging Spectroradiometer) retrieval and the polarimetric retrieval from the Polarization and Directionality of Earth's Reflectances (POLDER) instrument. Further, simulated POLDER data were used by Shang et al. (2015) to investigate the influence of cloud sub-grid variability to the POLDER derived cloud effective radius and variance.

In our study, the wide-field and highly resolved 2-D measurements of specMACS are simulated based on a realistic field of shallow cumulus clouds as observed during the EUREC⁴A campaign. The cloud data were obtained from LES using the PALM model (Raasch and Schröter, 2001; Maronga et al., 2015, 2020). This allows to apply the stereographic reconstruction and the cloudbow algorithm to the simulated measurements and compare the results to the respective quantities determined by the model cloud field used for the simulations. Although it is well known that the signal of the cloudbow originates from single scattering and hence, is weighted by $\exp\left(-\tau\right)$ with $\tau$ being the optical thickness (Alexandrov et al., 2012a), we will show that it is not straightforward to obtain the corresponding true model quantities. Nevertheless, the accuracy of the retrievals can be assessed and possible error sources can be quantified. This allows a deeper understanding of (multi-angle polarized) observations and their importance for the characterization of the clouds' microphysics.

## 2    Cloud data from LES-simulations with PALM

To evaluate the accuracy of the retrieval algorithms, a one minute overflight of HALO over a LES-simulated shallow cumulus cloud field as frequently observed during the EUREC⁴A campaign (Bony et al., 2017) in the vicinity of Barbados in early 2020 was simulated. Highly resolved LES-simulations were performed using the PALM model (Raasch and Schröter, 2001; Maronga et al., 2015). Within the $60\,\mathrm{s}$ overflight, the field of view of a single specMACS camera covers an approximate area of $32 \times 21\,\mathrm{km}^2$. Hence, a large cloud field of $25.6 \times 12.8\,\mathrm{km}^2$ horizontal extent was simulated for a duration of two minutes with

a second-by-second output at a horizontal grid size of $10 \times 10\,\mathrm{m}^2$ to match the high spatial resolution of the two polarization
cameras of specMACS. The vertical resolution was set to $5\,\mathrm{m}$ up to $2\,\mathrm{km}$ height, which is approximately the height of the
cloud tops. Above, the resolution is reduced until a resolution of approximately $15\,\mathrm{m}$ is reached at an altitude of $3\,\mathrm{km}$. The
LES-simulations were initialized by dropsonde measurements from 28 January 2020 during the EUREC[4]A campaign. On that
day, wide cloud patterns of shallow cumuli were observed (Stevens et al., 2021). In Fig. 1a, the vertical wind profiles of the
horizontal wind components $u$ and $v$ as well as the horizontal wind speed and its direction are shown for the first time step
used for the simulation of the specMACS measurement. Within the minute of the simulated overflight, the wind profiles do not
change significantly and hence, are representative for the horizontal movement of the clouds.

Following Maronga et al. (2015, 2020), PALM uses a bulk two-moment liquid-phase cloud microphysics scheme of Seifert
and Beheng (2001, 2006) providing cloud droplet number concentration ($N$) and specific water content (LWC). In our setup,
an extended scheme following Seifert and Beheng (2006), Khairoutdinov and Kogan (2000), Khvorostyanov and Curry (2006)
and Morrison and Grabowski (2007) was used. For the MYSTIC-simulations, the cloud microphysics need to be described in
terms of the liquid water content (LWC) and the effective radius ($r_{\mathrm{eff}}$) as well as the effective variance ($v_{\mathrm{eff}}$). While the LWC is
directly retrieved from the LES model output, the $r_{\mathrm{eff}}$ is derived from the model variables following Martin et al. (1994):

$$r_{\mathrm{eff}} = \left( \frac{3 \cdot \mathrm{LWC}}{4\pi \cdot k \cdot N \cdot \rho} \right)^{1/3} \cdot 10^{-6} \tag{1}$$

Here, $N$ is the water droplet density in $\mathrm{m}^{-3}$ and comes from the LES model output and $\rho$ is the water density ($1000\,\mathrm{kg\,m}^{-3}$).
$k$ is the ratio between the volume mean radius $r_v = \left( \int n(r) r^3 dr / N \right)^{1/3}$ and the effective radius each to the third power:
$k = r_v^3 / r_{\mathrm{eff}}^3$. For maritime airmasses, Martin et al. (1994) determined $k = 0.80 \pm 0.07$, hence $k = 0.80$ was chosen for the
calculation of the effective radius from the LES data following Eq. (1). The resulting distribution of effective radii for the first
simulated time can be seen in Fig. 1b. For all other times, the distribution looks similar (not shown here). For the radiative
transfer simulations, we assumed a constant effective variance of $v_{\mathrm{eff}} = 0.1$ to determine the optical properties of the clouds.
Since $k$ and $v_{\mathrm{eff}}$ are related by $k = (1 - v_{\mathrm{eff}})(1 - 2v_{\mathrm{eff}})$ for modified gamma distributions (e.g. Grosvenor et al., 2018) this
corresponds to a value of $k = 0.72$. However, as $k$ is only used to derive an effective radius which is not provided by the LES
model, the simulations are internally consistent and the choice of $k$ does not impact the analysis of this paper.

As shown by Marshak et al. (1998) for marine stratocumulus clouds, the radiative effects of a cloud are sufficiently well
represented in 3-D radiative transfer models if the spatial resolution of the model input resolves the mean free photon path $l$
of the clouds which is given by the inverse of the extinction coefficient $l = k_{\mathrm{ext}}^{-1}$. For the clouds obtained from the LES model,
the mean free photon path was roughly estimated to be on the order of $20\,\mathrm{m}$ which is comparable to the $20$–$30\,\mathrm{m}$ stated by
Marshak et al. (1998) for overcast marine stratocumulus clouds. Therefore, it was decided to reduce the horizontal resolution of
the grid for the computationally expensive radiative transfer simulations by a factor of two such that the grid-size is $20 \times 20\,\mathrm{m}^2$
while the vertical resolution was reduced by a factor of five to about $25\,\mathrm{m}$. In spite of the eventual resolution reduction for the
radiative transfer simulations, the highly resolved LES simulations with the horizontal grid size of $10\,\mathrm{m}$ remain crucial due to
the internal smoothing in the model.

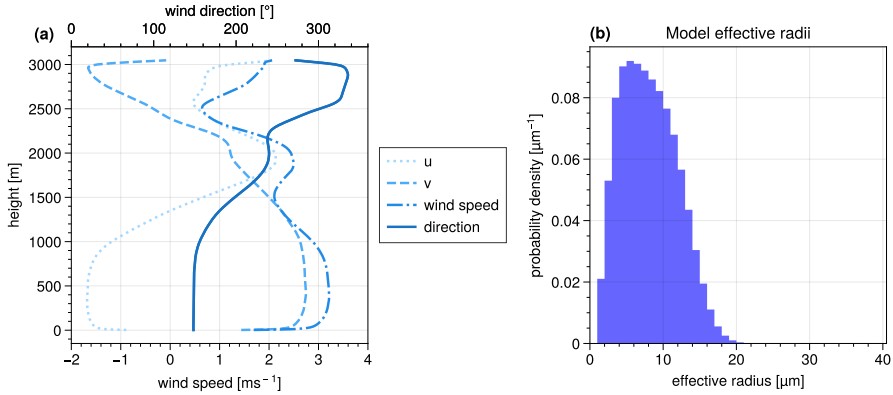

**Figure 1.** (a) Profiles of the horizontal wind vector components $u$ and $v$ as well as total wind speed (lower x-axis) and direction (upper x-axis). (b) Probability density distribution of effective radii in the model domain of the first simulated time.

## 3 3D radiative transfer simulations

The simulations of realistic measurements of the two polarization cameras of specMACS were performed using the 3-D radiative transfer model MYSTIC (Mayer, 2009) which is part of the freely available libRadtran radiative transfer package (Mayer and Kylling, 2005; Emde et al., 2016). MYSTIC allows the simulation of scalar radiances, and also of polarized radiation originating from scattering events of photons on cloud and aerosol particles or molecules (Emde et al., 2010). The number of photons was chosen so that the noise of the Monte Carlo simulated intensity had an average standard deviation of about $6\%$. The high computational costs were reduced by simulating measurements at a frequency of $1\,\mathrm{Hz}$ instead of the operating acquisition frequency of $8\,\mathrm{Hz}$ of specMACS. This is in accordance with the actual framerate at which the stereographic reconstruction algorithm is applied. While the operational cloudbow retrieval is carried out at an angular resolution of $0.3\,^\circ$, this is reduced to approximately $1.25\,^\circ$ for the simulations. Such a reduced angular resolution is still sufficient for the cloudbow retrieval: As shown in Fig. 13b in Miller et al. (2018), the minimum required Nyquist resolution to resolve the peaks of the supernumerary bows is about $1.2\,^\circ$ for the largest effective radius simulated here of about $40\,\mu\mathrm{m}$ and a wavelength of $0.49\,\mu\mathrm{m}$, and hence just below the given angular resolution. For longer wavelengths and smaller effective radii the minimum required resolution increases. However, as the effective radii in the model domain are mostly well below $25\,\mu\mathrm{m}$ (compare Fig. 1) at which the minimum required Nyquist frequency is about $1.5\,^\circ$ at a wavelength of $0.49\,\mu\mathrm{m}$, the features of the supernumerary bows, which are needed for an accurate determination of cloud droplet size distributions, are still well resolved in the simulations.

As described by Pörtge et al. (2023), the sensors of the two polarization cameras of specMACS are divided into $512 \times 612$ $4 \times 4$ pixel blocks for the different color channels (red–green–green–blue) and the polarization directions. For the whole field of view, the measurements are interpolated to the full $2048 \times 2448$ grid. Hence, to simulate the specMACS measurements, the Stokes vectors measured at each $4 \times 4$ pixel block are simulated and then interpolated to the whole grid with regard to the polarization directions and the spectral response of the different color channels. Hereby, the simulations are performed in the

wavelength range between $380\,\mathrm{nm}$ and $690\,\mathrm{nm}$ in $10\,\mathrm{nm}$ steps to represent the spectral response of the cameras as shown in Fig. 1b of Pörtge et al. (2023). The central wavelengths (bandwidths) of the three color channels are approximately $620\,\mathrm{nm}$ ($66\,\mathrm{nm}$), $546\,\mathrm{nm}$ ($117\,\mathrm{nm}$) and $468\,\mathrm{nm}$ ($82\,\mathrm{nm}$) (Pörtge et al., 2023). To represent the specMACS measurements, the simulations are weighted with the corresponding spectral response functions of the different color channels and polarization directions.

It was chosen to simulate a scene at a solar zenith angle of $30\,^\circ$ and a solar azimuth angle of $20\,^\circ$ such that the cloudbow was covered by the field of view of the camera. The flight direction was chosen toward the North with a horizontal aircraft attitude (the three Euler angles roll, pitch and yaw are $0\,^\circ$) for the whole flight. This geometry assured that the cloudbow is well visible in the field of view of the camera such that multiple cloud targets are observed from all scattering angles between $135\,^\circ$ and $165\,^\circ$ as required for the cloudbow retrieval (Pörtge et al., 2023).

Finally, MYSTIC allows to simulate complex fields of both liquid water and ice clouds (Mayer and Kylling, 2005; Mayer, 2009). In this study, the simulated shallow LES clouds consist of liquid water droplets only. The optical properties are calculated by Mie theory (Wiscombe, 1980).

## 4 Comparison of simulations to measurements

To begin with, it will be shown that the simulated cloud field is representative for shallow cumuli measured during the EUREC[4]A campaign. This will be demonstrated considering a scene measured on 28 January 2020 at 16:29 UTC by the so-called polLR camera (polarization camera looking to the lower right of the aircraft in flight direction (Weber et al., 2023)) of specMACS. The measured and simulated RGB images can be seen on the left of Fig. 2. Both scenes show shallow cumuli and from visual comparisons of the two RGB images it can already be stated that the simulations (middle panel) seem to be realistic. The simulations resolve optically thin and small clouds as well as the general cloud structure recognizable e.g. by shadows on the cloud surfaces. Shadows can also be detected on the underlying ocean surface, which is simulated using the bidirectional reflectance distribution function (BRDF) after Cox and Munk (1954a, b). In particular, shadows on the ocean surface are identifiable in the sunglint region, the specular reflection of the sun on the ocean surface. The position of the sunglint is determined by the relative position of the sun which is different in both images because the simulations were not conducted for a specific flight geometry. Still, simulation and measurement can be compared in terms of the radiances as the solar zenith angle (SZA) is comparable for the measurement (SZA $\approx 33\,^\circ$) and the simulation (SZA $= 30\,^\circ$).

For a more quantitative comparison, the measured and simulated radiances for the considered images are shown for the three color channels (red, green and blue) on the right of Fig. 2. Because of the similarity of the probability density functions of the different channels for the simulations and the measurements, the simulated cloud field can be stated to be representative for clouds as measured during the EUREC[4]A campaign.

In a similar manner, the components $Q$ and $U$ of the Stokes vector can be compared. For both, the simulations and the measurements, the Stokes vector is defined with respect to the scattering plane, such that $U \approx 0$ and the comparison can be reduced to the $Q$-component. The respective measured and simulated results for the green channel are shown in Fig. 3. From the spatial distributions of $Q$ it can be seen that the important features like the cloudbow and the sunglint show a linearly

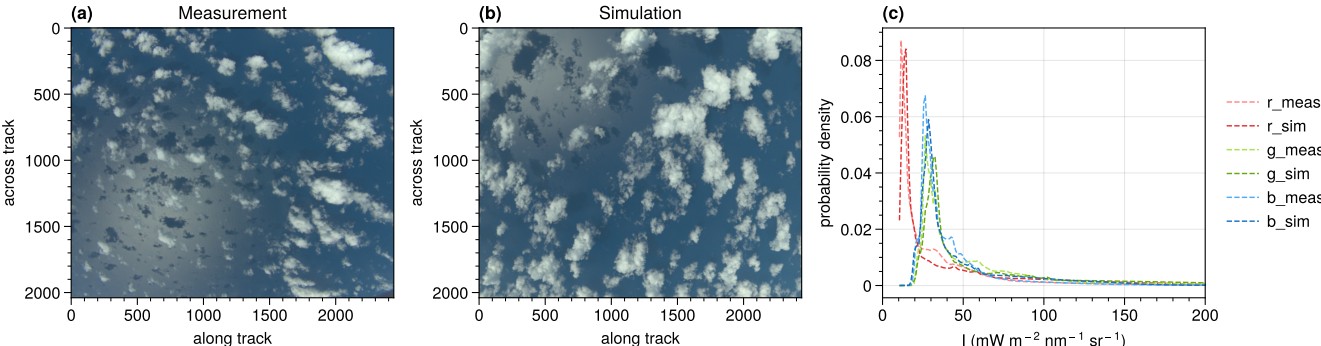

**Figure 2.** RGB images of the scene measured by the polLR camera on 28 January 2020 at 16:29 UTC (a) which is compared to the simulated cloud field 30 s after the simulation start (b). In (c), the radiance histograms of the scene measured by the polLR camera compared to the simulated cloud field are shown. The three color channels red (r), green (g) and blue (b) are shown separately for both the simulations (sim) and measurements (meas).

polarized signal of the same order for the measurement and the simulation. Since polarized radiances are much more sensitive to viewing geometry, the histogram at the right of Fig. 3 is restricted to the cloudbow region between $135°$ and $165°$ scattering angles. Except for some minor differences which are most likely due to the different geometry and cloud field considered, the histograms for the different channels look very similar. Hence, it can be stated that the simulations are representative for the polarized measurements as taken during the EUREC[4]A campaign.

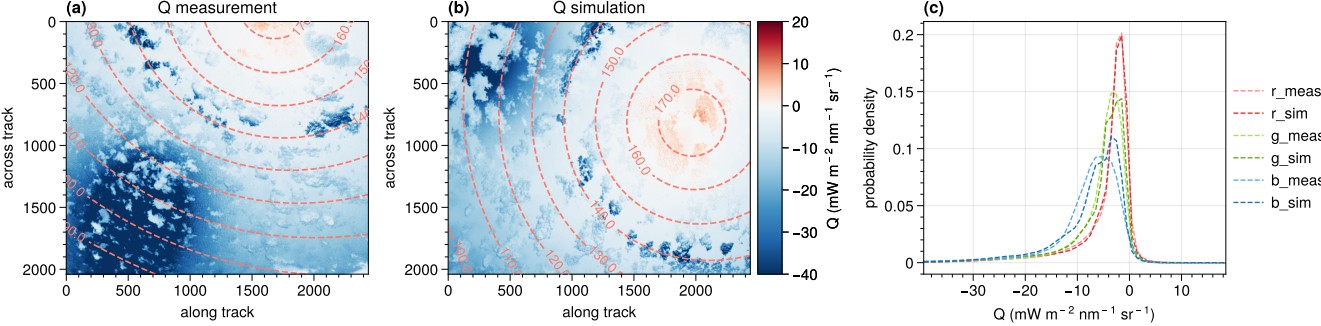

**Figure 3.** $Q$-component of the Stokes vector for the scene measured by the green channel of the polLR camera (a) which is compared to the green channel of the simulated $Q$-component 30 s after the simulation start (b). The dashed lines denote the scattering angles. In (c), the simulated probability density functions of the $Q$-components of the Stokes vector are compared for the cloudbow region between scattering angles of $135°$ and $165°$ to the measured ones. The three color channels red (r), green (g) and blue (b) are shown separately for both the simulations (sim) and measurements (meas).

## 5 Comparison of retrieval results to model data

A main question for the evaluation of the retrieval results is how the derived data like cloud top heights and effective radius can be compared to the actual model input. This is related to the question where the photons detected by the instrument originate from, and hence, which cloud top heights and cloud droplet size distributions we can expect to see from the model input. The polarized signal of the cloudbow is generated both by single and forward-directed multiple scattering, but since the latter does not affect the angular structure of the cloudbow it also does not affect droplet size retrievals (Alexandrov et al., 2012a). Hence, the relevant contribution to the measured signal comes from singly scattered photons which occurs on average after the mean free photon path (roughly $20\,\mathrm{m}$ as stated above) at an optical depth of $\tau = 1$. To account for this, we performed a second simulation with MYSTIC using the same viewing geometry as $30\,\mathrm{s}$ after simulation start and considered only singly scattered photons. For this simulation, the backward mode of MYSTIC was used, which implies that the photons are started at the detector and not at the sun (Mayer, 2009). It should be noted that the choice of a particular time and the corresponding fixed viewing geometry means that photons of oblique-viewing pixels might be scattered at higher altitudes in the cloud than nadir-looking pixels. However, we compared the retrieval results to two other reference simulations after $15\,\mathrm{s}$ and $45\,\mathrm{s}$ and verified that the comparisons are similar both from the individual simulations as well as a combination of the three reference simulations such that we will limit ourselves to the $30\,\mathrm{s}$ reference simulation in this paper. Performing those reference simulations without any scattering from molecules or aerosol ensures that the first scatter events of the photons will occur either on cloud droplets or the Earth's surface. From the scatter locations the indices of the respective grid boxes of the model input can be directly determined. On average, the first scattering events will happen at an optical depth of $\tau = 1$. We further used only a single wavelength, namely $550\,\mathrm{nm}$ corresponding approximately to the central wavelength of the green channel of the polarization cameras. Due to the similar penetration depth of the visible wavelengths, using a single wavelength is representative for the determination of the first scatter events. For each viewing direction, 1000 photons were simulated and their scatter event locations determine a weighting function along the line of sight of the instrument. Then, the average of all scattering event locations gives the location of the cloud part which is expected to be seen by the instrument, and hence, the expected cloud top height can be estimated. Although the cloud top height retrieval is based on intensity measurements and hence on the detection of both singly and multiply scattered photons, it is particularly important for the correct geolocalization of the cloud targets and the aggregation of the signal in the polarimetric retrieval. Therefore, the retrieved height should be as close as possible to the height from which the polarized and thus the single scattering signal originates. Moreover, the stereographic algorithm is based on the identification of contrasts gradients which are not visible deeper in the cloud. If an optical depth of $\tau = 1$ is not reached along the viewing direction, a corresponding fraction of photons will be scattered at the ground such that the vertical coordinate of the scatter event is zero. Those scatter events are not taken into account for the determination of the average scatter location. To compare the derived cloud droplet size distribution to the expected one from the model, all scatter event locations of photons that did not hit the ground are taken into account separately. This will be explained in more detail at the beginning of Sect. 5.2.

## 5.1 Stereographic reconstruction of cloud geometry

To begin with, the stereographic derived cloud top heights will be compared to the expected heights from the model. As already explained, the results of the retrieval algorithms were compared to the cloud data after a simulated flight time of $30\,\mathrm{s}$ corresponding to half of the totally simulated period. The corresponding RGB image and $Q$-component of the Stokes vector can be seen in Fig. 2 and Fig. 3. To compare the heights derived from the retrieval and the model, the points were projected to the pixels of the camera such that they can be compared point-wise. From the expected model data only those points were

selected where stereo points were derived and vice versa. The resulting cloud top heights as well as the respective distributions derived from the model input and the stereographic reconstruction are shown together with their point-wise differences in Fig. 4 for both the realistic simulation and the simulation where the clouds did not evolve in time. The derived points were projected to the same image as shown in Fig. 2b.

Figure 4 shows the heights from the model input and the stereographic reconstruction. They compare very well not only from their absolute values defined by the colors but also from their spatial structure. Comparing the histograms for the realistic

simulation on the left of Fig. 4, the stereographic derived cloud top heights have a mean of about $1575\,\mathrm{m}$ (panel d) while the model heights have a mean of about $1637\,\mathrm{m}$ (panel b). Hence, the stereographic derived heights are on average underestimated by about $62\,\mathrm{m}$ compared to the expected heights from the model input.

On the bottom of Fig. 4, the differences between the stereo heights and the model heights (panel e) as well as the corre-

270 sponding histogram (panel f) are shown. Areas where the stereographic reconstruction algorithm overestimates the cloud top heights are red, while blue areas mark underestimated heights by the stereo algorithm. The distribution of the histogram in panel f bears resemblance to a normal distribution with a shift to the left which shows the underestimation of the stereographic derived heights with a mean difference of about $\mu = -62\mathrm{m}$. The grey shaded area marks the $\pm\sigma$-interval with $\sigma \approx 130\mathrm{m}$ being the standard deviation. Moreover, as can be seen both from the histogram as well as the scatter points there are some significant

differences between single points up to $\pm500\mathrm{m}$ and more. The colors of the projected points on the bottom left of Fig. 4 show positive differences (red) mainly where the cloud is generally lower (i.e. at the edges and in the shadows) and negative (blue) values mostly at the cloud tops.

One possible error source of the stereographic reconstruction is the development of the clouds with time and their advection. As can be seen from Fig. 1a, the $v$-component of the wind vector is on average positive over the whole model domain and in all

280 heights, although its values range only between approximately $0.5\,\mathrm{ms}^{-1}$ and $2.6\,\mathrm{ms}^{-1}$. The flight direction was northward and due to the positive $v$-component, the velocity of the cloud relative to the moving airplane is reduced. This in turn is interpreted by the algorithm as lower cloud top heights. In order to eliminate this source of error and to estimate how the cloud development impacts the cloud top heights derived from the stereographic reconstruction, a simulation with the same geometric settings as before was performed. However, now the cloud input file given to MYSTIC was not changed while the aircraft was simulated

to fly over the model domain. Hence, the clouds do not develop and in this simulation we see exactly the same clouds from different perspectives during the overflight. Moreover, they do not move with the wind as they remain at the same position in the model frame for the whole simulation. The corresponding results from this simulation are shown on the right of Fig. 4. The

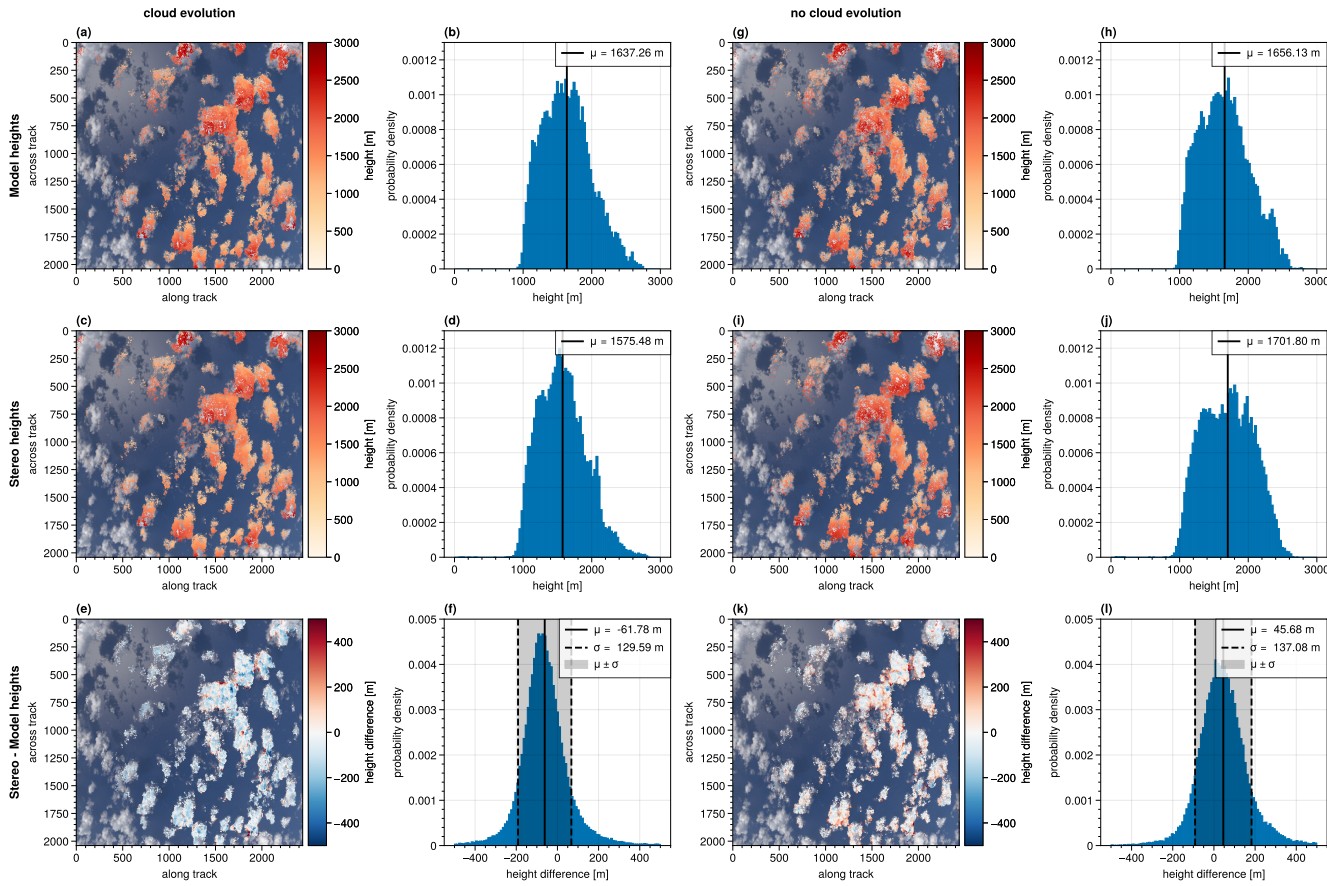

**Figure 4.** Comparison of the cloud top heights expected from the model and derived from the stereographic reconstruction algorithm for the two simulations with (left) and without cloud development (right). The cloud top heights as expected from the model input (Model heights) can be seen for the realistic simulation on the top left (panels a and b) and the simulation without cloud development on the top right (panel g and h). The stereographic derived cloud top heights for the two simulations are shown in the middle panel (panels c and d respectively i and j). Below, the point-wise differences are shown (panels e and f respectively k and l). The derived points were projected onto the simulated RGB image and the corresponding histograms are shown.

heights retrieved from the LES model input for the simulation without cloud development are shown on the top right of Fig. 4 (panel g and h). From the histogram, a mean value of about $1656\,\mathrm{m}$ can be extracted. The corresponding cloud top heights from the stereographic reconstruction method are depicted below (panel i and j). Here, the distribution indicates a mean value of $1701\,\mathrm{m}$. Comparing the two mean values shows an overestimation of the stereographic heights by approximately $46\,\mathrm{m}$.

Fig. 4k shows the point-wise differences between the stereographic retrieved cloud top heights and the model heights for the simulation without cloud evolution and in Fig. 4l the respective histogram of the differences can be seen. Next to the mean difference of about $\mu = 46\,\mathrm{m}$, a standard deviation of about $137\,\mathrm{m}$ is derived. Looking at the points projected to the RGB

image, the red and blue areas demonstrate again where the stereo heights are over- or underestimated compared to the model heights. Once more, the red areas are mostly located close to the cloud edges and in the shadow regions, while the blue areas are mainly in the middle of the clouds. Hence, this effect seems to be systematic and cannot be explained by the cloud evolution. One possible explanation for that could be the comparison of the stereographic cloud top heights, which are based on intensity measurements, to the model cloud top heights which were determined by only taking the first scatter events of photons started at the detector into account. This could lead to biases in the "expected" model heights when for example multiple scattering becomes important. However, as explained in Sec. 5, the algorithm is based on the identification of contrasts, which will not be visible deeper into the cloud. Moreover, the signal will smooth out when multiple scattering becomes more important making it harder for the algorithm to detect any features. The impact of multiple scattering on the contrasts detectable by the algorithm will be further addressed in future studies.

Nevertheless, it can be concluded that the cloud geometry and in particular the cloud top heights can be well determined using the stereographic reconstruction algorithm, with average differences to the expected heights of less than $70\,\text{m}$ and a standard deviation of about $130\,\text{m}$. The average deviation is expected to reduce even further if the wind movement of the clouds was considered. The small uncertainty in the cloud top heights found here is particularly valuable for the correct aggregation of the cloudbow signal and hence, the polarimetric retrieval of the cloud droplet size distribution. Because of the comparison to the expected model heights derived from the single scattering simulations, the uncertainty found here corresponds to the uncertainty in the origin of the polarization signal and hence the uncertainty for the signal aggregation in the polarimetric retrieval. Kölling et al. (2019) compared the retrieved cloud top heights to the cloud top heights derived from the WALES lidar for measurements during the NAWDEX campaign in October 2016 and found a median difference of $126\,\text{m}$. Hereby, the stereo heights were found to be lower. Furthermore, it was indicated that the most prominent outliers in regions of high lidar cloud top height and low stereo height were observed for thin cirrus layers above cumulus clouds. In those scenes, the lidar is sensitive to the upper ice cloud layer while the stereo algorithm detects image areas with high contrasts which are preferably observed for lower cloud layers. To overcome this problem of different instrument sensitivities, the realistic simulations performed in this study could be used to evaluate the performance of the stereographic reconstruction method. Similar stereographic techniques are also used for the derivation of cloud top heights from other air- and spaceborne instruments. For example, Moroney et al. (2002) find a predefined accuracy of $\pm 562\,\text{m}$ for the operational retrieval of the MISR (Multi-angle Imaging Spectrometer) cloud top heights derived on a $1.1\,\text{km}$ grid from aboard NASA's Terra satellite. This is explained by limitations in the matching algorithms of the stereo images. However, using an advanced sub-pixel least squares matching technique, this error could be reduced to $280\,\text{m}$ as shown by Seiz et al. (2006). Further, Seiz et al. (2006) applied the same algorithm to the Advanced Spaceborne Thermal Emission and Reflection Radiometer (ASTER) which is also operated on board of Terra and has a resolution of $15\,\text{m}$ which is comparable to the one of specMACS. For the ASTER cloud top heights, an accuracy of $12.5\,\text{m}$ is found with an additional uncertainty of about $100\,\text{m}$ for every $1\,\text{ms}^{-1}$ uncertainty in the wind component aligned with the direction of the satellite's orbital track. From RSP measurements cloud top heights are derived using the cross correlations between a set of consecutive nadir reflectances and sets at other viewing angles for multiple assumed cloud top heights (Sinclair et al., 2017). Then, cloud layers are identified from distinct peaks in the resulting correlation profile. Sinclair et al. (2017) find that the cloud top heights

derived from RSP have median errors of $0.5\,\mathrm{km}$ when compared to lidar measurements. Hence, the given accuracies for the cloud top heights are much higher than the ones found for specMACS in this work and the lidar-based comparison shown in Kölling et al. (2019). As for specMACS, the retrieved cloud top heights are used as input for the aggregation process RSP measurements (Alexandrov et al., 2016). The analysis in this paper shows that the cloud top heights derived from specMACS using the stereographic approach have much smaller uncertainties and hence, should lead to smaller errors in the aggregation of the polarized radiance signal on which the polarimetric technique relies.

## 5.2 Cloud droplet size distributions

As described in Sec. 5, the locations of the cloud top height and effective radii seen by the instrument can be determined from simulations of singly scattered photons. From the scatter locations, the model grid boxes in which the scatter events occur are determined and hence, the effective radius of the cloud droplet size distribution at which the photon is scattered. Since photons detected by single pixels of the detector are scattered in various model grid boxes, the signal that is actually seen by that pixel originates from different cloud droplet size distributions. For real measurements, the actual distributions are not known but the cloudbow retrieval assumes that the signal comes from cloud droplets obeying a modified gamma distribution as described by Hansen (1971) with

$$n(r) = n_0 r^{(1-3b)/b} e^{-[r/(ab)]}. \tag{2}$$

The two parameters $a := r_\mathrm{eff}$ and $b := v_\mathrm{eff}$ are referred to as the mean effective radius $r_\mathrm{eff}$ and the effective variance $v_\mathrm{eff}$ respectively which can be used to define the radiative properties representative for a cloud. They are given by

$$r_\mathrm{eff} = \frac{\int_0^\infty r \pi r^2 n(r) dr}{\int_0^\infty \pi r^2 n(r) dr} \tag{3}$$

and

$$v_\mathrm{eff} = \frac{\int_0^\infty (r - r_\mathrm{eff})^2 \pi r^2 n(r) dr}{r_\mathrm{eff}^2 \int_0^\infty \pi r^2 n(r) dr}. \tag{4}$$

In this study, the cloud properties were explicitly defined for the simulations: The droplet size distribution of each grid box is given by a modified gamma distribution with a constant effective variance of $v_\mathrm{eff} = 0.1$ and the respective effective radius of the grid box. As described above, the signal measured by single pixels of the detector originates from scattering events in different grid boxes and thus, different droplet size distributions. Hence, to find the effective radius and variances which are really seen by the instrument, the gamma distributions of the single grid boxes in which the photons are scattered have to be superimposed. As shown by Shang et al. (2015), the sum of two or more gamma distributions is not another gamma distribution and in particular are the cloud droplet effective radius and variance of the combined gamma distributions not just the averages of the respective quantities of the single distributions. The grid box of the model data (and therefore the corresponding droplet size distribution) is determined from the position of the singly scattered photons. For all the pixels which are simultaneously evaluated by the cloudbow retrieval, the distributions seen by the singly scattered photons of the corresponding simulated

pixels can then be superimposed to obtain one distribution as seen by the instrument. The resulting distribution does not have a precisely determinable shape but the effective radius and variance of that distribution can be calculated using two different ways: At first, the cloudbow retrieval assumes a gamma distribution for the droplet size distribution so it would be reasonable if the derived effective radius and variance resembles the effective radius and variance of the best fitting modified gamma distribution to the total droplet size distribution. Second, the effective radius and variance can be calculated using their definitions after Eq. (3) and Eq. (4). For modified gamma distributions in the form of Eq. (2) and a constant effective variance for all sub-distributions, this can be simplified as shown by Alexandrov et al. (2012a), such that

$$r_{\text{eff}}^{\text{tot}} = \frac{\langle r_{\text{eff}}^3 \rangle}{\langle r_{\text{eff}}^2 \rangle} \tag{5}$$

and

$$v_{\text{eff}}^{\text{tot}} = v_{\text{eff}} + \left( \frac{\langle r_{\text{eff}}^4 \rangle \langle r_{\text{eff}}^2 \rangle}{\langle r_{\text{eff}}^3 \rangle^2} - 1 \right) (1 + v_{\text{eff}}). \tag{6}$$

$r_{\text{eff}}^{\text{tot}}$ and $v_{\text{eff}}^{\text{tot}}$ denote the total effective radius and variance of the combined distributions while $r_{\text{eff}}$ and $v_{\text{eff}}$ are the respective quantities of the single distributions. The angular brackets denote averages. Hence, the effective radius and variance that should be derived from the polarized measurements of the cloudbow are $r_{\text{eff}}^{\text{tot}}$ and $v_{\text{eff}}^{\text{tot}}$ respectively.

Both methods showed nearly identical results for the calculation of the expected effective radius and variance from the model input for the simulations of this study. Therefore, in the following, we will compare the results of the cloudbow retrieval to the best fitting gamma distribution of the single-scattering simulations only. For all figures the same time point as for the stereo heights was chosen (30 s after simulation start) and only those parts of the image where the cloudbow retrieval could be applied will be compared.

In Fig. 5 the expected effective radius from the model input is shown and compared to the results from the cloudbow retrieval for both the realistic simulation with developing clouds (left) and the idealistic case where the clouds did not evolve over time (right). Compared to the stereographic derived cloud top heights considered before, it can be seen that only parts of the image are evaluated. This is due to the specific scattering angle range ($135°$ to $165°$) under which a cloud target needs to be observed during the overflight such that the cloudbow algorithm can be applied. Moreover, the $10 \times 10$ pixel cloud targets have a coarser resolution than the points found by the stereo tracker for a cloud scene as considered here, where many contrast gradients are identified by the tracking algorithm. To start the comparison between the cloudbow results and the expected model input, we observe that the spatial distributions of the projected effective radii on the RGB image derived from the cloudbow retrieval closely match the spatial distributions expected from the model input. This holds true for both cases of evolving and non-evolving clouds. Moreover, the histograms show a similar width of the distributions of effective radii. For the case of evolving clouds, the mean effective radius for the scene derived by the cloudbow retrieval is $10.32\,\mu\text{m}$ and compares well to the expected mean effective radius of $10.49\,\mu\text{m}$ from the model input. In case of non-evolving clouds, the mean deviation from the cloudbow retrieval to the expected effective radii reduces slightly with expected mean effective radii of $10.60\,\mu\text{m}$ compared to $10.57\,\mu\text{m}$ derived by the cloudbow retrieval.

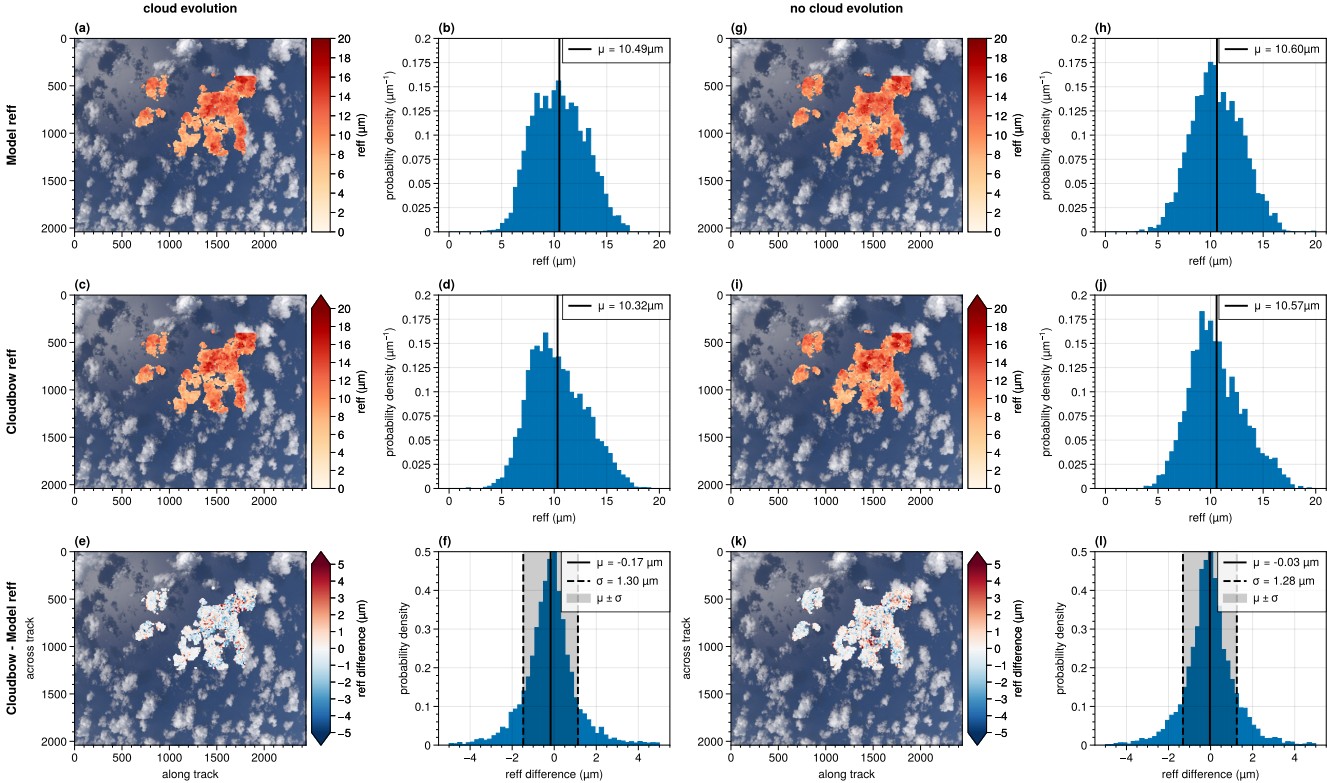

**Figure 5.** Comparison of the cloudbow retrieval result for the effective radius (middle) expected model effective radii (top) for the simulation with cloud evolution (left, panels a to f) and the simulation where the clouds did not develop (right, panels g to l). On the bottom, the corresponding point-wise differences are shown. The points are projected to the RGB image of the simulation and next to that the respective distributions are shown in form of a histogram with the mean value marked by the solid vertical line. The grey shaded area in the lower histograms marks the $1\sigma$-interval.

On the bottom of Fig. 5 the point-wise differences between the effective radii from the cloudbow retrieval and expected ones from the model are shown as well as the corresponding histograms. As before for the cloud top heights, red areas mark overestimations and blue areas underestimations of the effective radius by the cloudbow retrieval compared to the expected effective radii from the model input. For the case of developing clouds, the spatial distribution of the differences projected to the RGB image shows rather blue values, thus indicating an average underestimation of the effective radius by the retrieval. A mean difference between the retrieval results and the model of $-0.17\,\mu$m is derived. Moreover, the standard deviation is given by $1.30\,\mu$m. For the idealistic simulation of non-evolving clouds, the mean difference reduces to $-0.03\,\mu$m with a standard deviation of $1.28\,\mu$m. For the clouds and the viewing geometry considered in the simulation, it takes about $35\,$s to observe one cloud target under the necessary scattering angles between $135\,^{\circ}$ and $165\,^{\circ}$. During that time, the clouds develop in the case of the realistic simulation hence influencing the multiangular measurement of the polarized radiance. Moreover, the clouds move

at a speed of $2$–$3\,\mathrm{m\,s^{-1}}$ as can be seen from Fig. 1a for the determined typical cloud top heights between $1000\,\mathrm{m}$ and $2000\,\mathrm{m}$. This corresponds to horizontal displacements of $70\,\mathrm{m}$ to $105\,\mathrm{m}$ within an observation period of $35\,\mathrm{s}$ and hence, displacements on the order of one cloud target. Consequently, the retrieval becomes even more accurate when considering clouds that do not

develop and do not move with the wind.

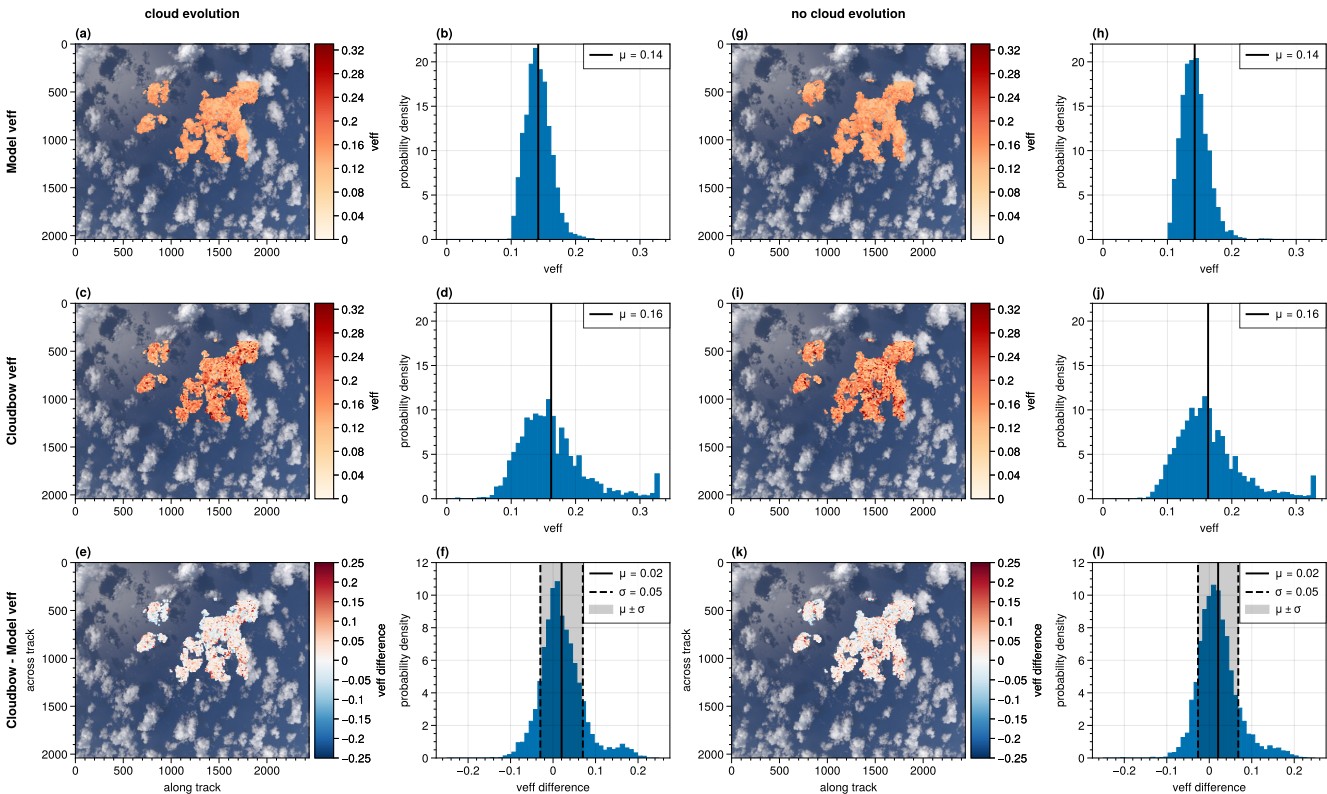

**Figure 6.** Same as Fig. 5 but with the expected and derived effective variances.

A similar analysis can be done for the results of the effective variance which are given in Fig. 6. Again, the results are given for the two simulations (evolving and non-evolving clouds) although they differ hardly from each other. For both simulations, the expected mean effective variance is given by $0.14$ with a rather narrow distribution (compare Fig. 6b and h). It should be noted again, that there is no underlying signal in the effective variance itself as it was held constant over the model domain

with a value of $v_{\mathrm{eff}} = 0.1$. Nevertheless, the distributions show that all expected effective variances are larger than $0.1$, which shows the variation in the effective radius within one cloud target and is as expected from Eq. (6). In future studies, additional variations in the effective variance should be considered such that the sensitivity of the polarimetric retrieval on natural variations in the effective variance is also tested. A comparison of the cloudbow retrieval results to the expected ones shows wider distributions and a larger mean of $0.16$ for the two simulations. The mean difference between the cloudbow retrieval and the

expected ones from the model is $0.02$ with a standard deviation of $0.05$. The distribution of the cloudbow retrieval shows an

accumulation of effective variances at 0.32 which is the maximum effective variance covered by the retrieval. To summarize, the difference distributions on the bottom of Fig. 6 show a general overestimation of the effective variance by the cloudbow retrieval (mostly red points in the spatial distribution). This is supported by the histograms in panels f and l.

We further studied the correlations between errors in the effective radius and variance and the underlying optical thickness for all target points. The results are shown in Fig. 7 as normalized probability densities. From Fig. 7a which shows the retrieved effective variance with respect to the retrieved effective radius, it can be seen that large effective variances occur over the full range of retrieved effective radii. The accumulation of effective variances at the maximum effective variance covered by the retrieval is also observed in retrievals applied to real measurement data (Pörtge et al., 2023). Therefore, we performed further studies with a focus on retrievals with $v_{\text{eff}} = 0.32$ by looking at correlations to other parameters given either by the cloud field studied (i.e. optical thickness) or derived from the retrieval, i.e. the effective radius and its difference to the expected one or the goodness of the fit. However, none of the performed analyses showed a clear correlation. Moreover, we studied potential correlations between the effective radius and variance errors which can be seen in Fig. 7b. It can be seen that most retrievals have small differences in both the effective radius and variance, which emphasizes again the general accuracy of the retrieval. However, there are no significant correlations between large effective radii differences and large effective variance differences. The two lower panels of Fig. 7 show the correlations between the two differences to the nadir optical thickness of the observed cloud target. From both panels c and d it can be seen that most of the points have low optical thicknesses. In Shang et al. (2016) it was shown that polarized reflectances (which are proportional to polarized radiances for constant solar zenith angles) not fully saturate for optical thicknesses smaller than $\tau = 10$. This in turn could be thought of impacting the results of the retrieval. However, as can be concluded from the same figures, there are many points which show accurate results also for low optical thicknesses. Besides, there are no further obvious correlations between errors in the effective radius or variance and the optical thickness.

An explanation for the overestimation of the effective variance derived by the cloudbow retrieval might be the large sub-grid variability of the signal (not shown here), which has been observed during the evaluation process for the shallow cumulus clouds considered in this study. A similar sub-grid variability could also be observed for measurements of highly structured cloud fields and hence might indicate that the application of the retrieval on those cloud types can be difficult. In particular are large effective variances retrieved if the supernumerary bows of the cloudbow are suppressed (Pörtge et al., 2023) which might be the case for highly variable signals. Likewise, Shang et al. (2015) observed a high biased effective variance for inhomogeneous cloud fields on the sub-grid scale.

A systematic bias towards higher effective variances has also been observed by Alexandrov et al. (2012a), who evaluated the performance of the RSP retrieval algorithm on realistic clouds. Tests on 1-D plane parallel simulations in the solar principle plane revealed deviations for the effective variance between $6\%$ and $27\%$ and decreasing errors towards smaller $r_{\text{eff}}$, while the retrieved effective radius showed on average an accuracy better than $0.15\,\mu\text{m}$. Alexandrov et al. (2012a) explained the bias in the effective variance with the "smoothing" effect on the polarized reflectance generated by multiple scattering. Moreover, they concluded from comparisons between 3-D and 1-D simulations that vertical variations in the effective radius are interpreted as larger effective variances in the 3-D simulations compared to the 1-D analogues due to the side illuminations of the clouds.

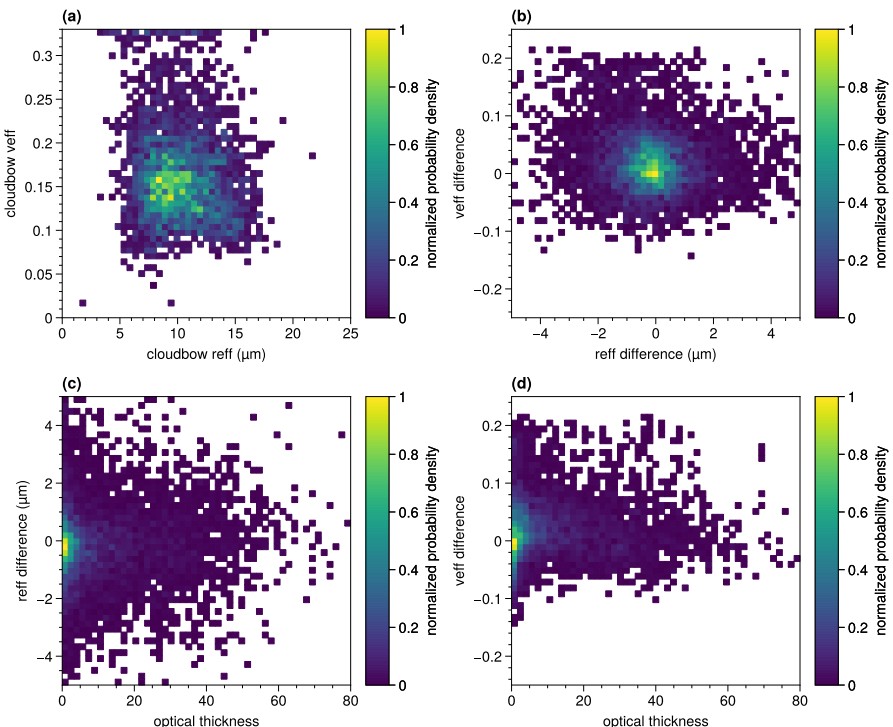

**Figure 7.** Correlations between retrieved effective radius and variance (a), the corresponding differences to the expected values from the model in (b), and the relations of the effective radius (c) and variance (d) differences to the cloud optical thickness.

Hence, the results from Alexandrov et al. (2012a) in combination with the observed large sub-grid variability in the polarized radiance for highly structured clouds can explain the bias towards higher effective variances derived from the cloudbow retrieval than expected from the model input.

Fig. 8 shows an example of the expected cloud droplet size distribution as derived from the model input for one $10 \times 10$ pixel cloud target of the realistic simulation. The corresponding gamma distribution fit used for the comparison to the cloudbow retrieval result is shown in blue with an effective radius of $r_{\text{eff}} = 13.22\,\mu\text{m}$ and an effective variance of $0.12$. The cloud droplet size distribution associated with the effective radius of $r_{\text{eff}} = 12.95\,\mu\text{m}$ and an effective variance of $0.15$ derived from the cloudbow retrieval for that target is shown in red dashed. Despite the underestimation of the effective radius by $-0.27\,\mu\text{m}$ and the overestimation of the effective variance by $0.03$, the derived cloud droplet size distribution is still very similar to the one expected from the model input. Hence, being able to determine the effective radius and variance with the average accuracy found in this study, shows that the actual cloud droplet size distribution can be well retrieved from polarized measurements of the cloudbow in case the droplet sizes follow a simple gamma distribution. For arbitrary distributions, the so-called Rainbow Fourier Transform (RFT) described by Alexandrov et al. (2012b) allows to retrieve the actual shape of the distribution.

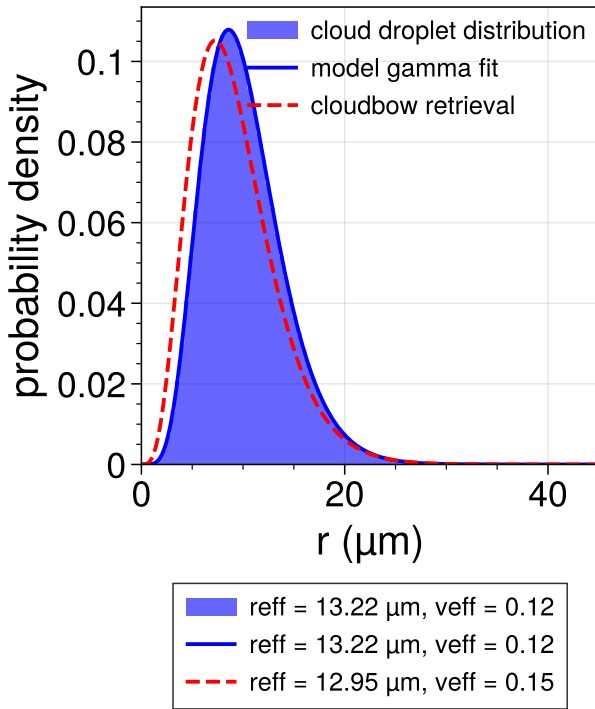

**Figure 8.** Probability density distribution of cloud droplet radii determined from the LES model input with respect to one target pixel of the cloudbow retrieval. The blue line gives the corresponding best gamma distribution fit. In red, the cloud droplet size distribution as derived for the cloud target from the effective radius and variance determined by the cloudbow retrieval is shown.

To summarize, we found that the accuracy of the effective radius is on average $(-0.17 \pm 1.30)\,\mu$m. Compared to the average
effective radius of $10.49\,\mu$m expected from the model input, this corresponds to a relative error of less than $2\,\%$. For the effective variance, we found an average accuracy of $(0.02 \pm 0.05)$. The average expected effective variance from the model input was $0.14$ such that the corresponding error is less than $15\,\%$. The presented results are similar to the ones found by Alexandrov et al. (2012a) for simulations of the 1-D along track RSP measurements at a wavelength of $865\,$nm. As stated above, specMACS delivers 2-D measurements and has broader channels which might influence the accuracy of the polarimetric retrieval. Here we
showed that the spectral response is not a severe limitation for the clouds considered and it can be stated that the effective radius and the effective variance can be retrieved with a high accuracy using measurements of the polarized radiance from specMACS in the region of the cloudbow. In particular, Mishchenko et al. (2004) described the requirements on measurements of the size distribution parameters of liquid water clouds for the quantification of aerosol effects on climate and stated an accuracy better than $1\,\mu$m or $10\,\%$ for the effective radius and $0.05$ or $50\,\%$ for the effective variance. Taking this into account shows even
more how valuable the measurements of the size distribution parameters derived from the angular shape and structure of the cloudbow can be for climate research.

## 6 Conclusions

In this study, measurements of the polarization resolving polLR camera of specMACS flown on board of the German research aircraft HALO were simulated using the 3-D radiative transfer model MYSTIC. The aim was the analysis of the accuracies of the retrieval algorithms applied to the measurements of the camera. The first algorithm is the stereographic reconstruction algorithm to determine the cloud geometry and cloud top heights. The second algorithm is the polarimetric retrieval which uses polarized measurements of the cloudbow to derive microphysical properties of the cloud droplet size distribution. Both algorithms rely on the observation of clouds from multiple viewing angles. Hence, a one minute overflight over a realistic cloud field obtained from simulations with the LES-model PALM was performed using the 3-D radiative transfer model MYSTIC. The LES-simulations were initiated based on dropsonde measurements from the EUREC[4]A field campaign which took place in early 2020 in the vicinity of Barbados studying shallow cumulus convection. In that way, it was ensured that the cloud field represents shallow cumuli as measured during the campaign which could be shown by a qualitative comparison of the measured and simulated (total and polarized) radiance. In addition to the realistic simulation with advective and developing clouds, a theoretical simulation of non-evolving clouds was performed to test the sensitivity of the retrieval algorithms to cloud development and movement.

For the stereographic reconstruction of the clouds, it was shown that the derived cloud top heights differ on average by less than $70\,\mathrm{m}$ from the expected model heights with a standard deviation of about $130\,\mathrm{m}$. In the case of non-evolving clouds, the deviation from the expected heights reduced to about $46\,\mathrm{m}$ with a standard deviation of about $137\,\mathrm{m}$. A comparison of the point-wise differences revealed that the stereographic cloud top heights tend to be underestimated at the highest cloud tops, while they tend to be overestimated where the clouds are lower, i.e. at the cloud edges or in shadows.

Next to the cloud top heights from the stereographic reconstruction, the cloud droplet size distributions obtained in form of the effective radius and effective variance from the polarized measurements of the cloudbow were tested in this study. Here, it was shown that the effective radius from the retrieval differs on average by about $(-0.17 \pm 1.30)\,\mathrm{\mu m}$ from the expected effective radius of the model input. In the case of non-evolving clouds the average difference is $(-0.03 \pm 1.28)\,\mathrm{\mu m}$, and thus slightly better than the result for the realistic simulation. The results for the effective variance were shown to be very similar for both simulated cases with evolving and non-evolving clouds. The mean difference was $0.02$ with a standard deviation of $0.05$. The distribution of effective variances derived from the cloudbow retrieval is broader than the expected distribution. The wider distribution might be explained by the fact that the effective variance highly depends on the signal of the supernumerary bows. If that signal is damped, larger effective variances are retrieved. The large sub-grid variability which has been observed for the shallow cumulus clouds considered in this study might cause such a damping of the signal. Comparing the results of the realistic simulation including the cloud development and the idealistic case without any cloud evolution indicates, that the cloud development might also be one source of error for the overestimation of the effective variance by the retrieval. Furthermore, a "smoothing" of the signal due to contributions from multiple scattering and the stratification of the effective radius within the cloud can be reasons for this overestimation as already observed by Alexandrov et al. (2012a). With respect to some outliers with larger errors in the effective radius or variance, we studied the correlation between the differences in the effective radius

and variance retrieved and expected from the model and the optical thickness. However, no significant correlation was found which could point to an error source.

In future, further investigations of the observed sub-grid variability are planned. In particular, a reduction of the resolution of the LES model output could be used to exclude any sub-grid variability. Moreover, it is planned to include natural variations in the effective variance in future studies. Furthermore, the approach of 3-D radiative transfer simulations will be used for the accuracy assessment of the two retrieval algorithms for other cloud types such as deeper convective or mixed phase clouds.

*Data availability.* The PALM-LES simulations are published by Jakub and Volkmer (2023). The specMACS data used in this study are available upon request from the corresponding author.

*Author contributions.* LV performed the 3D radiative transfer simulations with MYSTIC, applied the stereographic reconstruction algorithm to the synthetic data and evaluated the results. VP and BM helped setting up the radiative transfer simulations and actively participated in discussing the results. Furthermore, VP applied the cloudbow retrieval to the simulated data. FJ performed the required highly-resolved model simulations using the PALM model.

*Competing interests.* At least one of the (co-)authors is a member of the editorial board of Atmospheric Measurement Techniques.

*Acknowledgements.* We want to thank Tobias Zinner and Anna Weber for their contributions in the general discussions during the working process. Further, the authors want to thank Claudia Emde for helping with questions regarding polarization. This research has been supported by the Deutsche Forschungsgemeinschaft (DFG, German Research Foundation) within the Priority Project SPP 1294 and the Grant SFB/TRR165 ("Waves to Weather").

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
