# Peer review of "Model-based evaluation of cloud geometry and droplet size retrievals from 2-D polarized measurements of specMACS"

_EGUsphere, 2023_

## Author Comment (AC1)

**Response to RC1 of egusphere-2023-2235**

We want to thank the first referee of our paper for the review and comments. One major point of the review was the similarity of the results to the previously published results by Alexandrov et al. 2012, who studied the accuracy of the polarimetric cloud droplet size distribution retrieval applied to simulated measurements of the RSP instrument. To begin with, we think that it is a good result that we independently found a high accuracy of the polarimetric retrieval (for a different instrument) and that this also affirms the results of Alexandrov et al. 2012. Moreover, we want to highlight why our study is substantially different from the one published by Alexandrov et al. although the same retrieval approach is addressed:

- The specMACS measurements differ from the RSP measurements as we obtain 2-D polarized measurements which allows the retrieval of cloud droplets size distributions on 2-D images rather than in the principal plane only. The 2-D measurements also allow an accurate determination of cloud top heights from the stereographic retrieval which is also addressed in this work. Further, the accurate determination of the cloud top heights is important for the polarimetric retrieval and is considered in this work by evaluating the accuracy of the full retrieval procedure used for real measurements. This includes the initial identification of potential cloud targets, the geolocalization including the stereographic cloud top heights and the fit of the polarized phase functions to the aggregated signal. On the contrary, using RSP measurements, the simulated radiances can be aggregated to a defined point on the cloud surface. With regard to Fig. 15 in the paper, this is expressed by Alexandrov et al. as follows: "The horizontal position of a 'pixel' used for the comparison between 1D and 3D retrievals is depicted by the dashed line. The RSP reflectance simulated with the 3D RT model output was aggregated to the point where the dashed line crosses the cloud top boundary."
- Providing 2-D measurements comes with the cost of broader spectral channels (three RGB channels), which might result in "smoothed" cloudbow signals and hence, the need to study the accuracy of the polarimetric retrieval applied to specMACS measurements. Therefore, obtaining similar results to the ones of Alexandrov et al. 2012 confirms that accurate cloud droplet size distributions can also be obtained from less spectrally resolved cameras.
- The MYSTIC simulations performed for Alexandrov et al. include only a single wavelength and did not consider the spectral response function of the RSP instrument.
- The clouds did not develop over time in their simulations while we also studied the impact of cloud evolution on the cloud top height and reff/veff retrievals.
- In Alexandrov et al. only a single point is compared to the effective radius expected from the LES input data. The statistics performed are mainly for potential biases in 1-D cloud assumptions.

The listed points are now addressed in the paper from line 97 onwards and are again discussed in line 467f.

General comments:

1. The results are largely consistent with those of Alexandrov et al. 2012, who used essentially the same retrieval approach and the same 3D radiative transfer model. In that sense the current paper does not add so much to the available literature. I would suggest to add a slightly deeper analysis of the cases where the retrievals substantially deviate from the truth. Are cases with large errors in Reff and Veff related? Do they correspond to low optical thickness cases? Do these retrievals exhibit a bad fit of the simulations to the data? Is there any way such cases could be identified in case of real retrievals?

   In addition to the above explained differences between the work of Alexandrov et al. 2012 and ours, we included a study of the retrieval results that substantially deviate from the truth in the paper. The results are shown in Fig. 7. Here, we show the correlations between the retrieved effective radius and variance (panel a), their differences (panel b) and the correlation between the differences and the vertical optical thickness of the observed cloud targets. However, there are no significant correlations between the different quantities, as explained in the paper between lines 419 and 436.

2. The paper refers to earlier paper for the details about the drop size distribution retrievals. However, I think it is crucial information to understand this paper and suggest that a short description of the retrieval procedure is included.

   We added a short description from line 69 onward.

3. In line 114 it is stated that k = 0.8 is assumed for the calculations of effective radius from the assumed number concentrations. However, in line 116 it is stated that an effective variance (veff) of 0.1 is assumed. This is inconsistent, as k and veff are related by k = (1-veff)*(1-2*veff) for the gamma distributions assumed here (see for example Grosvenor et al. 2020 Eq. 13; https://doi.org/10.1029/2017RG000593). For an assumed veff=0.1, k would be 0.72. I do not think this matters much for the analysis in the paper, so I will not ask to correct this. It should be noted in the paper, however.

   Yes, thanks for noticing. We noted the inconsistency from line 150 onward and explained why this should not affect the results of the paper.

4. It is not clear to me if "evolution" of the cloud also includes movement of these clouds. If so, please make this clear and mention the windspeed (profile) assumed in the paper and also indicate how much the clouds generally move during the cloudbow observations.

   The realistic simulation with cloud "evolution" includes the movement of the clouds within the LES model domain. We added the associated horizontal wind profile in Fig. 1. We used the wind profile to explain why an underestimation of the cloud top heights in

the stereographic retrieval is possible (l. 278f.) and calculated the horizontal displacement of the cloud targets during the cloudbow observations in l. 399f. to 70-105m.

Specific comments:

Line 133: For clarity, please change to "acquisition time of 8Hz of SpecMACS."

We added "of specMACS" to the manuscript, but did not change "frequency" to "time".

Line 206: A flight time of 30 seconds is simulated. Please specify what area is covered by all viewing angles in such a time.

Thanks, our description was probably not clear enough. We simulated a total flight time of 60s (see Sect. 2) and tried to clarify this as well in line 255f. We added the information that a single camera of specMACS covers an approximate area of 32 x 21 km$^2$ within the simulated time over the LES field in l. 128.

Line 252: The accuracy of stereo heights retrievals can be compared to those estimated by Sinclair et al. 2017 (https://doi.org/10.5194/amt-10-2361-2017), who analyzed data with many different cloud types.

Thanks for the advice. We compared our results to the results of Sinclair et al. 2017 and further to the stereographic cloud top height retrievals for MISR and ASTER described by Moroney et al. 2002 and Seiz et al. 2006 from l. 318f.

Line 284: If the optical path length for each viewing angle is similar, won't different viewing angles see slightly different physical vertical locations in the cloud top? Then each angle may see slightly different size distributions. I guess this effect is minor, but it might be good to estimate this. Also, which viewing angle is taken into account for the analysis here?

With the setup for the single scattering (reference) simulation, we get indeed only a particular viewing angle for the comparison. However, as explained now in Sec. 5, we performed two more reference simulations at different times (after 15 and 45s). The corresponding viewing angle of a specific cloud target is then also changed. We verified that the overall results obtained for all the reference simulations and their combination are similar. By comparing the results for different cloud targets, we found that the expected effective radius can deviate by 1-2μm between the different reference simulations with different viewing angles, but the total results hardly change.

Line 332: Please give the mean and standard deviation of the differences here. They are only given in the conclusions.

Thanks for noticing. We included the mean and standard deviations as suggested.

Figure 4, 5 and 6. In the scatterplots all the low values are hard to see. Please change the colorbar.

Since the second referee suggested to present the histograms of the differences instead of the scatterplots, we decided to do that instead of changing the colorbar as suggested here.

---

## Author Comment (AC2)

**Response to RC2 of egusphere-2023-2235**

First, we want to thank the second referee for reviewing our manuscript. In the following, we will respond to the given suggestions and questions and describe the changes made within the manuscript corresponding to the comments in detail.

General Comments:

1. The authors make the claim that they are "validating" retrieval results in several places in the introduction. Given the way they are comparing "model" and "retrieval" results I am not sure that is entirely the case, as there is no evaluation of whether polarized reflectance misfits are related to "errors" in retrieved properties. There is also no evaluation of the effects of noise on the retrievals. Nonetheless the use of a simple model sampling scheme to relate observations to model variables is of considerable value for understanding how to use polarized observations for model evaluation and should probably be more emphasized.

Thank you for noting the perhaps misleading use of the term "validate" in the submitted paper. We changed the affected formulations to weaker terms, for example "evaluation of the accuracy of the retrievals". The arguments of the referee that there is no full validation of the retrievals are understandable. But in fact, measurement noise was considered in our evaluation: As explained in Sec. 3, the noise of the Monte Carlo simulations was set to 6%. This is slightly higher but on the order of the recently found total radiometric uncertainty for the two polarization cameras of specMACS of 3.8% to 5.8% (see Weber et al, https://egusphere.copernicus.org/preprints/2023/egusphere-2023-2209/egusphere-2023-2209.pdf) and could of course lead to higher variations in the aggregated polarized radiance signal. We also simulated a subset of the image for the full overflight with a smaller noise of 3% previously to the submission of this manuscript, and did not find significantly different results for the retrievals. In addition, the comparison between the real measurements and the simulations in Sec. 4 shows that the simulations performed are realistic and thus, they were treated as if they were real measurements. We emphasized the point mentioned in this comment at the end of the Introduction (l. 118f.).

2. The reverse Monte-Carlo sampling of singly scattered photons described at the beginning of Section 5 is the crux of the paper and presents a reasonable and simple way of sampling model output, in order to evaluate it against polarimetric observations. However, the rationale for the use of "the average of all scattering event locations" (line 199) in sampling the model fields and its consequences should be discussed. For example, a "model" simple average, unweighted by the probability of the path will tend to give a sample that is biased deeper into the cloud than that of the optical signal if singly scattered light dominates. In contrast, if multiply scattered light

(e.g. multiple forward scattering events caused by the diffraction peak in the phase function) dominates the signal the "model" simple average will give a sample that is biased higher in the cloud than the optical signal. This difference between the "model" sampling and what one expects to be the source of the optical signal is one potential explanation for the compensating biases in the stereo cloud top height results. Some discussion of the single scattering "model" versus full Monte Carlo sampling would therefore be helpful.

The authors agree with the argument given by the referee that taking an unweighted average of the scatter event locations for the determination of the "expected" model quantity results in biases with respect to the actually observed quantities. From the backward Monte Carlo simulations of singly scattered photons, we get the locations of the last scatter events of photons on the way from the sun to the detector weighted with $\exp(-\tau)$ along the line of sight. Hence, the "average of all scattering event locations" is not an unweighted average. Nevertheless, one can discuss about the influence of multiple scattering on the optical signal detected at the sensor and thus seen by the retrieval algorithms and how its neglect will bias the expected model quantities. To begin with, the polarized signal of the cloudbow is generated by both single and multiple scattering, the latter, however, has no significant influence on the angular structure of the cloudbow and hence the droplet size distribution retrieval (see Sec. 3 and Alexandrov et al. (2012)). Therefore, it is reasonable to consider the distributions seen by singly scattered photons. In contrast, the stereographic retrieval uses simple intensity measurements, such that multiple scattering might dominate the signal. However, it is based on the identification of contrast gradients. Those will decrease significantly, when multiple scattering becomes more important as the signal will be smoothed. We plan to study this in future by addressing the influence of the different scattering orders on the contrasts of the image. We noted this as a potential bias in the manuscript in Sec. 5 (l. 241f.) and discussed it further in l. 298f. Moreover, the stereographic cloud top heights are used for the geolocalization of the cloud targets evaluated by the cloudbow retrieval. Hereby, a small error in the cloud top height can already affect the aggregation of the cloudbow signal significantly as explained by Pörtge et al. (2023) as well as in the Introduction (l.70f.) and in Sec. 5. Thus, the comparison of the stereo heights to the expected ones from the single scattering simulations gives valuable information about the uncertainty of the height information used for the cloudbow retrieval. We pointed this out in line 299 onward.

3. Figures 4f & l, 5f and l and 6f and l are not useful. The statistics of the regressions are informative but for the reader presenting histograms of differences would be a more effective use of the graphic.
   The suggestion made by the reviewer to present the difference histograms was implemented in the manuscript including the mean and the 1σ-interval.

4. As noted in point 2 there will be a difference in the vertical weighting of the "model" sample and that which is expected from the polarimetric retrievals. It is particularly important to note this when making a comparison between the effective variance

retrievals and the model since the effective variance itself is constant. This means that there is no underlying signal, and the comparison is primarily a comparison of differences in sampling. An additional point regarding the effective variance comparison is that the retrievals have clearly failed when veff=0.32. Some additional comment on this and ideally examining whether there are a particular range of effective radii where this failure occurs would be desirable.

Correct, because of the constant effective variance throughout the model domain, there is no natural variation in the effective variance and we have not tested the retrieval on variations in the effective variance itself for now. For that, a parametrization to derive the effective variance from the outcome of the LES model would be needed. We plan to include this in future studies. As explained in l. 408f now, the variation in the effective variance which is currently expected from the model input is only due to variations in the effective radius within the sampling volume.

Concerning the cases where veff=0.32, we performed a slightly deeper analysis, however, as pointed out from l. 423 onward, no significant correlation between retrieved effective variances of 0.32 and other parameters was found. Concerning the question of the range of effective radii, it can be said that veff=0.32 occurs for nearly all effective radii retrieved as can be seen from Fig. 7a.

5. While I do not think that additional simulations are in order the authors should note that an effective variance of 0.1 is quite large for a cloud top size distribution and this should be born in mind when planning future work.

Thank you for pointing this out, we will certainly keep that in mind. For the EUREC4A campaign, effective variances on the order of 0.1 were found. Since we are continuing these model-based evaluations, we will use observed veff distributions in future studies.

Editorial Suggestions

Rewrite Eq.(6) as v_eff_tot= veff + (reff_4*reff_2/reff_3^2-1)*(1+veff) where reff_4 is the 4th moment of the effective radius etc. I suggest this because it is then clear that sampling variability in reff can only increase the apparent effective variance.

Where the Marshak et al. (1998) paper is cited at line 121 it should be noted that there conclusions are for overcast clouds. E.g. insert the word overcast between "for" and "marine" on that line.

Both editorial suggestions were applied to the manuscript.